# The Use of Geonarratives to Add Context to Fine Scale Geospatial Research

**DOI:** 10.3390/ijerph16030515

**Published:** 2019-02-12

**Authors:** Jayakrishnan Ajayakumar, Andrew Curtis, Steve Smith, Jacqueline Curtis

**Affiliations:** 1Department of Geography, GIS, Health & Hazards Lab, Kent State University, Kent, OH 44240, USA; acurti13@kent.edu (A.C.); jcurti21@kent.edu (J.C.); 2Department of Social Science, Missouri Southern State University, Joplin, MO 64801, USA; Smith-Steve@mssu.edu

**Keywords:** Spatial Video, Geonarratives, GIS, Geo-computation

## Abstract

There has been a move towards using mixed method approaches in geospatial research to gain context in understanding health related social patterns and processes. The central premise is that official data is often too reductionist and misses’ nuances that can help explain causality. One example is the geonarrative, a spatially relevant commentary or interview that can be mapped by content and/or location. While there have been several examples of geonarratives being used by researchers, there is no commonly available software that can easily transfer the associated text into spatial data. Having a standardized software platform is vital if these methods are to be used across different disciplines. This paper presents an overview of a solution, Wordmapper (WM), which is a standalone software developed to process geonarratives from a transcription and associated global positioning system (GPS) path. Apart from querying textual narrative data, Wordmapper facilitates qualitative coding which could be used to extract latent contextual information from the narratives. In order to improve interoperability, Wordmapper provides spatialized narrative data in formats, such as ESRI shape files, Keyhole Markup Language (KML), and Comma Separated Values (CSV). A case study based on five different spatial video geonarratives (SVG) collected to assess the human impacts following the 2011 Joplin, Missouri are used for illustration.

## 1. Introduction

In 2018 a series of natural disasters, such as the Paradise wildfire in California, have devastated communities in the United States. The recovery process from such events begins with the initial search and rescue, and then extends through multiple years of physical and social rebuilding. Along that continuum, the human toll is immense—from the trauma of the initial experience through various psychopathological manifestation during recovery. These events are spatial, temporal and human. To fully address such dynamic landscapes, we need nuanced data that combines more traditional surveillance with ground-up insights. Indeed, the addition of context into geospatial health research is imperative if researchers are to understand the process, as well as outcome [1] addressing what Kwan describes as “*the uncertain geographic context problem*” [2,3]. Environmentally inspired narratives, or geonarratives, provide a new means to collect spatially anchored insight [4,5,6,7]. The rationale is compelling, have someone who knows about/is impacted by an event talk about what has happened/is happening. These insights can then be used to enrich other more traditional data layers. In this way data gaps can be identified, while more traditional spatial analytical output can be contextualized. For most geonarratives, a commentary is recorded while moving through an environment. The addition of video, as is used in a spatial video geonarrative (SVG), captures even more context, in effect allowing the researcher to virtually recreate the narrative to *see* where comments were made and what is being described [8].

There are many advantages to this form of data collection, especially for challenging environments. While the video provides a visual resource that can be used for digitizing map layers, the simultaneously collected global positioning path (GPS) path anchors each frame to an exact location. Onto these mapped layers, conceptually, are the comments that provide “depth” to the map. Again, each comment is matched to both location and video frame providing a rich understanding of the outcomes and processes at work. This same approach can then be applied across multiple interviews collected for the same time, or across different periods, to gain multiple perspectives of a location, or to see how that location changes. In this way more traditional data, such as overdose locations, can be enriched with an understanding that reaches beyond a point on the map. However, what has been missing to make SVG a truly ubiquitous tool, is an easy means to combine video, narrative and coordinates in such a way that a non-GIScientist can fully utilize this method. In this paper we address this deficiency using a research support tool called Wordmapper.

Mobile technologies provide exciting new avenues of spatially focused research across a variety of disciplines. These techniques provide a novel means to capture data for micro spaces of activity [9,10,11,12,13,14]. In the study of human agency, these technologies can also provide a means to add “context” to more traditional geospatial analysis [15]. While the importance of these technologies and approaches are widely appreciated [16], an impediment to their more common application remains the ease and utility of how to transform textual data with spatial references into mapped formats. Thus, some researchers who might utilize this approach remain unconvinced because of their lack of geospatial and programming skills. It is therefore important to provide a bridge from mobile spatial technologies and their data exported to a platform that qualitative researchers with diverse methodological backgrounds can easily use. In this paper, we show how this can be achieved by merging commentary and geography into simple, mapped outputs. To illustrate the potential of this approach we use a case study of geonarratives collected after the tornado that devastated Joplin, Missouri in 2011. The purpose of those narratives was to consider the human consequences along the devastation to recovery continuum.

Health researchers incorporating spatial data have argued that we need to consider-the-local because of its importance in designing effective intervention. Here we mean “local” as a subsection of a neighborhood, along a street, or even around a building. While new geospatial technologies can provide data layers to inform investigations at this scale, an important consideration is the addition of context; not just, *where* a hotspot occurs but *why*? One approach to a more nuanced consideration of spatial data is a qualitative geographic information systems (GIS) approach [17,18,19], which can include mixed methods [20], geovisualizations [21,22] and narratives [6,23,24]. One of the most exciting methods in terms of enriching context are “*Geonarratives*” [6], which use the functionality of the GIS to analyze and interpret narrative materials, such as interviews, oral histories, life histories and biographies. Madden and Ross [25] demonstrate how geospatial technologies can be used to support narrative testimonials of individuals related to human right issues, while Kwan illustrated how emotional geographies can be incorporated within geospatial technologies to provide a richer representation of the lived experience [26]. Curtis and colleagues extend this work by combining geonarratives with simultaneously collected video to provide visually and spatially supported context across a variety of different sub-neighborhood spaces [27,28]. In this last example, spatial video (SV), which are video encoded with, or combined with, a coordinate stream, provide a valuable addition to the narrative as the researcher can return to the image to see what and where was being discussed. SV is in itself a novel geospatial approach that can be used to create spatial layers for different data poor environments [29], with examples, including disaster science [30,31,32], medical and health geography [8,27], environmental and social justice [33], urban geography [34] and crime [31,35,36]. In these examples, the addition of a spatial video geonarrative (SVG) provides associated context through the day-to-day experiences and insights of those who occupy that same space.

While SVG offers an advantage to qualitative researchers with an interest in more explicitly spatial data, for example, why one corner of a city park is associated with violence, an obstacle has been the lack of any easy-to-use software. There are examples of customized software developed for research needs; Kwan and Ding’s [6] 3D-VQGIS could be used for the analysis of textual data in a GIS, while Mennis, Cao, and Mason [7] created a visualization system for the exploration of narrative activity data using ArcGIS and Visual Basic (VB). For SVG, Curtis et al. [28] utilized a customized web program called G-Code to create geo-tagged words that could then be mapped in a GIS. However, the implementation of all these approaches has been far from ubiquitous. With GIS becoming popular for application focused research in various domains, such as public health, anthropology, archaeology, psychology, economics, and political science [37], developing user-friendly tools for cross-domain research becomes more important [38]. In this paper, we bridge this gap between the research potential as identified by geospatial scientists, and the applied use of geonarratives by users with a limited geospatial skill set. In so doing we open this field to not only a broader set of topical domains, but also to the type of research needed on how variations in input, data collection, and method selection can affect geonarrative research.

## 2. Design and Components

To address this research gap for a more ubiquitous use of SVG we have developed a conceptual frame for how narratives can be mapped. This can be seen in Figure 1, which is comprised of six modules used to address the *geonarrative problem*. This means that transcribed text in the form of a narrative and associated GPS data needs to be combined to create a geonarrative dataset where comments and words become spatial objects. By doing this a researcher can generate queries based on textual attributes, such as keywords, and spatial queries based on the geotagged content.

To achieve this goal, a working conceptualization model based around six modules was developed. This schema provides the inspiration for Wordmapper. There now follows a more detailed conceptual explanation of these modules while also linking to the practical application as experienced when using Wordmapper.

### 2.1. Preprocessing Module

Once a narrative has been collected, it is then transcribed. In order to be able to match the text to a spatial location, the time when each comment or word occurs in the narrative must also be recorded. The preprocessing module accepts a narrative in the form of a text file and GPS data in the form of a Comma Separated Value (CSV) file (see the upcoming Figure 4A). The narrative is a text file with each sentence having the timestamp on the audio recording inserted at the beginning. The GPS data is a Comma Separated Value (CSV) file with latitude, longitude, and time in Universal Time Coordinated (UTC) format. The offset time between these two data inputs is the difference between the media time of the audio recording and GPS time. Simply put, if the audio starts at 0:01 seconds, and we know the Greenwich Mean Time (GMT) of the GPS when the first word is spoken, we can sync both data streams together.

Apart from accepting these inputs, the preprocessing module should also validate the data inputs, especially out of sync time stamps in the transcribed narrative. We used a custom GPS correction software [39], developed by the GIS Health & Hazards lab at Kent State University (Kent, OH, USA) to correct the positional errors in GPS. The corrected GPS is only used for further analysis

### 2.2. Combiner Module

The combiner module syncs the narratives and GPS data to create geonarratives. To do this all narrative sentences are initially of the tuple form <timestamp, sentence>, which along with offset time is used to match the starting index in the GPS data (Figure 2). The starting index in the GPS is utilized to generate geonarratives sentences of the tuple form <timestamp, sentence, location>. The mapped sentences are further utilized to generate words with spatial coordinates through a word interpolation algorithm.

The first step in generating spatial words consists of the tokenization of the sentence. Natural Language Toolkit (NLTK) [40], a Python-based text processing library, is used for a more sophisticated tokenization strategy. This is in preference to the commonly used white space based tokenization approach which can create meaningless word tokens that may degrade the quality of geotagged words. An illustrative example is of the tokenization of a sentence, “I…am here.” A simple white space based tokenization would split the sentence into two words, ‘I…am’, ‘here.’, which are not the intended word tokens. An NLTK based tokenizer would instead have as output five words, ‘I’, ‘…’, ‘am’, ‘here’, ‘.’, from which the meaningful three words can be easily extracted. The timestamp information from the narrative record is used to assign a timestamp for each word beginning a section (the narrative that follows the time stamp in the transcription) as a tuple of the form <word, timestamp>.

The GPS locations for the start and end of a narrative sentence provide coordinates for the first and last word respectively. From the coordinates for the first word (*X*_1_, *Y*_1_) and the last word (*X*_2_, *Y*_2_), the *angle of incidence* of the path (*α*) can be calculated using the *Point-Slope* equation. By assuming a constant speed of travel and an associated known time stamp, the distance between an unknown word and a reference location, *z*, could be found out using average speed formula. By utilizing the distance (*z*), and the *angle of incidence* (*α*), and a reference word location (*X*_1_, *Y*_1_), the coordinates for the new word (*X, Y*) can be calculated using trigonometry (Figure 3). Once the location information is deduced, the word can be represented as a triplet of the form <word, timestamp, location>. A maximum interpolation time can also be added in case there is too much “dead air” between comments. The narratives that are successfully processed and mapped are added to a narrative list for the user to conduct further exploratory analysis.

### 2.3. Visualization Module

The visualization module allows for an interactive dynamic visualization of geonarrative. The JavaScript-based Google Maps API is used to display the narrative path, as well as narrative sentences with coordinates as markers. Each marker also displays as a Google Maps Info window with the narrative text (Figure 4B). Apart from the interactive map, Wordmapper also has a table with each row displaying the narrative text which dynamically animates (Figure 4C) the corresponding marker, zooming into that particular location of the sentence.

To further aid the visualization of the spatial words and to help evolve the search through co-occurring words and phrases, a dynamic and interactive wordcloud (Figure 4D), which is a graphical representation of word frequency, is added. A further advantage of the wordcloud is that the searches (and visual representation) are matched with mapped output, which can help to identify where the mentions are made. In other words, this is a form of a spatialized wordcloud, which again promotes iterative searches.

### 2.4. Query Module

The query module supports extensive keyword-based searches (Figure 4E). Apart from supporting strict keyword matches, the query module also supports wildcard-based searches. For example, a search with the keyword ‘recover*’ can match geonarrative sentences with the word ‘recover’, ‘recovering’, ‘recovered’, and ‘recovers’. Again, as mentioned in the previous section, the wordcloud can also be used to generate keyword-based search through click interactions.

### 2.5. Category Module

Apart from spatio-temporal and contextual information, geonarratives are also an excellent source for extracting latent knowledge (Figure 4F). Therefore, having the ability to create category types, and then being able to assign each comment into those categories can help with both spatial and thematic investigations. For example, comments in a narrative could be assigned to different themes (health, violence, recovery etc.), or by spatial content (spatially specific, fuzzy or inspired mentioned), or even by time (far past, recent past, current).

### 2.6. Output Module

Even though the preceding modules are important in creating a data set that can be investigated, manipulated and even mapped within the same software, it is also important to have the flexibility to be able to transition these data to other software. For example, for those with GIS experience, the spatial narrative data, and the geotagged words are output as two types of ESRI shape files (Figure 4G). These include a point file where every word in the narrative has a location, and secondly a coordinate for the beginning point of every transcribed section of the narrative. These GIS outputs allow for the spatialized narratives to be combined or compared with other relevant datasets. For example, if the user had entered the search term “injury”, then the corresponding point shapefile has a column where every comment, and every word, are identified as matching the query. These can then be used as input to a kernel density analysis to create a hotspot map of injury perception [27]. The corresponding SVG comment shapefiles that fall inside these hotspots can then be read for additional context about what was said regarding various injuries at the time of the disaster and in the recovery period after. The same approach works with hotspot analyses of more traditional data, such as 911 calls for service, overlaid on the same damaged and recovering areas in the months after the disaster [27]. For researchers, not familiar with a GIS, Wordmapper provides an alternative output in the form of Google Earth’s spatial data format, Keyhole Markup Language (KML). The benefits of Google Earth are many, including ease of use, and free satellite imagery and aerial photos of most of the earth’s land surface making it a popular research tool for mapping in the health domain [41,42]. The KML output mirrors that for the GIS; a path of points with a visible word attached, and points showing the start of each narrative section. While not having the power of manipulation and analysis of a GIS, the user can still see exactly where words are located, can zoom into the map for a particular location, or use the text in the side bar content field to go to a key part of the narrative, and then read the longer text stream at and around that location. This makes these data spatially accessible and interpretable for any researcher. Of note is that any search word is immediately identified as a yellow pin on both the map and in the table of contents. This means a researcher with no geospatial technology training can still gain a geographic perspective on the narrative. Wordmapper also provides output for non-spatial textual analysis for specialist qualitative analysis software, such as NVivo.

To illustrate the potential of collecting SVG data and then analyzing it using Wordmapper, a small case study is presented based around the devastation of the 2011 Tornado in Joplin, Missouri. This has been chosen as events in the United States during 2017 and 2018 have left multiple communities devastated by flood, fire and hurricane, and the health impacts of those events will extend across a multiyear period in terms of direct and indirect disaster related illness. The SVG provides a way to capture this complex interaction of spatial, social and temporal outcomes.

## 3. An Empirical Illustration: A Spatial Video Geonarrative in Joplin, Missouri

SVG have been collected for a variety of different environments and topics, including the health needs of a community, predicting crime, homelessness, overdoses and risks associated with infectious disease. However, the first applications of both SV and SVG were on the physical damage and subsequent health consequences of post disaster landscapes [33,43,44,45]. Here we use the Joplin, Missouri, tornado of 2011, which was the deadliest single path tornado in the United States since 1947 with over 160 deaths, to showcase how SVG can be used to contextualize a dynamic landscape.

The tornado hit in the early evening of Sunday 22 May leaving a 6-mile path of destruction with widths of up to 1/2 mile. Multiple land use areas were destroyed, including several residential neighborhoods. Soon after the tornado a SV team mapped the damage [30], and then repeated this survey at regular intervals to monitor recovery. As part of this recovery research, several SVGs were also collected in 2014 to record different perspectives of the storm, the complexity of recovery, and especially the human perspective and challenges faced. An initial mapping of these SVG used a precursor to Wordmapper called G-Code that interpolated words between the beginning points of transcribed narrative sections [28]. While this was an important tool to begin the mapping of a SVG, the output consisted of an online text screen from which three columns of data (ID, time and word) had to be copied and pasted into Excel, and then manipulated in a GIS for mapping purposes. While some collaborators could work through these steps, for many it was a problem. A further limitation was that G-Code resided on a server and connectivity problems sometimes prevented access. Given these limitations, and with a growing desire to expand collaborations, the previously described way to standardize the mapping of a SVG was conceptualized, and Wordmapper software was developed.

Five SVG’s were collected during 2014. For each ride, two “Contour+2” cameras were mounted on a vehicle, while an audio recorder captured the commentary of a test subject as he/she navigated through what had been the tornado path. After collection, all narratives were transcribed, with each subject comment having a media time stamp in the form of [hh:mm:ss] added before the first word. The GPS path was extracted from the video (using Contour Storyteller software) as a CSV file, and the Greenwich Mean Time (GMT) of the video media time that matched the first word of the transcription was used as the offset time for Wordmapper.

The preprocessing module in Wordmapper reads in the narrative file, csv file with location, and offset time. The combiner module combines the narrative and location data to create geonarratives, which are added to the Wordmapper geonarrative dataset queue. The dynamic wordcloud generated from the Wordmapper provides an initial comparison of the narratives. Figure 5a displays a general wordcloud for one of the narratives without any keyword search. The wordcloud suggests that words, such as “tornado”, “apartment”, and “rebuilding” are frequently used in the narrative. A second query uses “recover*”, which matches “recover”, “recovers”, “recovering”, and “recovered”. The corresponding wordcloud (Figure 5b) indicates that ‘rebuilding’ is a high frequency word associated with ‘recover’. These provide an initial insight into what might be an interesting second search, such as “rebuilding” (Figure 5c).

The visualization window in Figure 6 displays the path associated with the keyword ‘recover*’. The red pin shows the location of all the narrative sentences while the yellow pin identifies those containing the searched for keyword.

Figure 7a shows one of the search results matching the word ‘recover*’ and indicates a question by the interviewer. By examining other matching sentences, we found that the word ‘recover’ is mainly used by the interviewer to enquire about the recovery process after the tornado to the subject. By examining the consecutive narrative sentences (Figure 7b), we found that the responses from the subject contains valuable insights about recovery without mentioning the word “recover” (rather words, such as ‘rebuild’, were used). Two things can be gleaned from this. Firstly, the everyday conversation about recovery may not actually include the term “recover”, and instead the activities and outcomes of “recovery” are described. Recovery may be an academic rather than a colloquial term. Thus, it would be useful to return to the comments proximate to the “recovery” cues to identify more appropriate search words. This may not have been so clear without the mapping approach. Indeed, this iterative exploratory approach is one of the most compelling features of Wordmapper.

In another example (Figure 8) an interviewee mentions a convenience store that was totally demolished by the tornado. The corresponding news extracted from the New York Daily News [46] confirms the validity of the statement. This example shows how external sources can be used to triangulate the findings.

It is also important to go beyond simply mapping text, and instead mine the content, and the structure of the content further. To do this, Wordmapper allows for the creation of different categories, which can then be used to create comment subsets. For example, while these categories could be thematic (recovery or mental health for example), they could also be spatial. Here we have created two categories ‘Location Specific’ and ‘Fuzzy Space’ which are used to classify comments that have explicit mention of a location, such as ‘this house’, ‘that church over there’, or comments that have vague place mentions, such as ‘this area’, and ‘that street’. A combined wordcloud (Figure 9) from both categories indicates the use of deictic words, such as ‘here’, ‘this’, ‘there’ and ‘that’. The spatial deictic words could further be used to aid Machine Learning based approaches that would automatically classify spatially cued sentences.

Additional investigative power occurs when these tools are combined, for example a search based on the keyword ‘damage’ and the category ‘Fuzzy Space’ helps to identify locations where mention about ‘damage’ was made. From the example (Figure 10), it seems that that the word ‘damage’ is more associated with an area than a particular location. A more specific location tends to have more detailed descriptions.

However, keywords can also be used as a point source for analysis, and to illustrate this, three different damage related categories, including, ‘Low Damage’, ‘Medium Damage’, and ‘High Damage’ were created using Wordmapper. Only narrative sentences that contain spatial information along with content related to damage were classified. A sentence, such as “*And this whole area, maybe a little wind damage, but that’s it*” was classified as ‘Low Damage’ and given the value 1, while a sentence, such as “*Moderate to light damage. I know that that particular church for a while was providing shelter to some people*.” was classified as ‘Medium Damage’ and given a value of 5, and sentence, such as “*Here ? that convenience store there was one of the places that was totally demolished, and I guess there was a recording of a cell phone call that people hiding in the ? people hiding in the freezer. got, uh, widely distributed nationally.*” was classified as ‘High Damage’ and was given a value of 10. Kernel Density Estimation, a technique to calculate the density of point and line features, was used to create a density surface with the classified damage values. The map (Figure 11) shows a high concentration of damage mentions near the intersection between South Rangeline Road and E20th Street and Rex Avenue. From newspaper articles, it could be identified that Walmart, Home Depot, and Academy Sports and Outdoors, which were in the intersection near to South Rangeline Rd and E20th Street, were completely destroyed in the tornado [47]. The second main hotspot in Rex Avenue is near the Plaza apartment, which was completely destroyed in the tornado [48].

While this type of categorical heat map may not replace actual damage estimates (in this example), but it does provide the basis for comparative mapping to see how this approach could be used in areas where no or little data exists. For example, the above heat map could be repeated for emotional content, or mentions of recovery related illnesses or other health conditions. In both examples maps made from more traditional sources are likely to be incomplete.

## 4. Discussion

Any spatial data, whether physical (locations of damage), or health (suicide locations) only tell a partial story unless contextualized. One way to capture these insights is through an SVG. Now, using Wordmapper, we also have the means to fully leverage these data. More specifically, Wordmapper allows for the transcribed narrative to be combined with coordinates in an easy to use interface that supports both textual and spatial investigations. These queries are not linear in the sense of a more typical qualitative GIS approach, but rather iterative, supported by images and text. The addition of categories, that can include both content themes and spatial structure, along with supporting visuals, such as maps and wordclouds, encourage the user to evolve their search in a nonlinear way. Wordmapper has been designed for use by non-spatial science, nor computer science savvy users. In this way a local mayor’s office, or a non-profit, or a public health official could benefit from SVG use. This has the additional benefit of also making these groups more willing collaborators as the benefits are tangible.

Wordmapper is designed to be standalone software, which both maximizes the security of working with potentially sensitive data [49,50], while limiting the reliance on access to an external server, which also may be prone to network outages, and server downtimes. These are important considerations for collaborators working in the field, especially in locations where internet connectivity has questionable security, for example, international work in data challenged locations. Even though the software can provide almost all functionalities, including search, download, and qualitative coding without an internet connection, the map display based on Google Maps API requires an internet connection. In future revisions of the software we plan to include data driven document (D3) maps with minimal mapping features as an alternative to Google Maps during internet connection disruptions. An important security concern regarding spatial confidentiality would be the usage of Google Maps API for narrative location display. As we are not transferring textual data across to Google servers, the confidentiality issue is minimal, even though we plan to incorporate more privacy preserving mapping approaches, such as offline mapping (loading map tiles) for future revisions of the software. Indeed, while the illustration used here involved a post-disaster landscape, the topical application of SVG is broad. Current SVG collaborations involve multiple overseas environments associated with infectious disease, or multiple cities in the United States addressing problems of homelessness or child injury, or even more rural communities fighting opioid addiction. The point is that SVG is a ubiquitous method that now has an equally accessible means to fully leverage these data.

Developing Wordmapper, however, is only the opening to many other strands of associated research on the technique itself. It has previously been suggested by Curtis and colleagues that comments are either spatially precise (for example “people were injured in this building here”), spatially fuzzy (“all around this area the sirens couldn’t be heard”) and spatially inspired (“I’ve heard a lot of people complaining about dust”) [4]. It would be interesting to see the proportion of comments falling into these categories, and then how each was cued in the narrative. Could, for example, certain key words be used to automatically extract out only those spatially precise comments, such as ‘deictic’ words, which could provide contextual information about a person, place, and time [51]. It would also be interesting to see how close each word describing a specific place is to that place, which would then have knock-on implications with different methods of analysis. Questions could also be investigated with regards to how such findings vary based on location, cohort or the framing influence of the interviewer. While there is a rich body of research into interview techniques, the SVG is a sufficiently different method to warrant further investigation, and Wordmapper can help with that analysis.

With advances in the area of Natural Language Processing (NLP), researchers are using approaches like sentiment analysis to identify and extract emotional context from sources, such as microblogs [52] web-forums, and narratives [53]. With the help of open source tools, such as Natural Language Toolkit for Python (NLTK), which already supports basic level sentiment analysis, textual data in the form of narratives could be used for the systematical empirical investigation of emotions [54]. It could be argued that SVGs offer a far richer data set than these other forms of text, and as they also contain the location, there is the possibility of conducting a place-based sentiment analysis.

## 5. Conclusions

Spatial video geonarratives are an exciting new approach for data collection. They can capture institutional knowledge, record a variety of different perspectives for the same space, be used for historical and contemporary investigations, reveal where overdoses are likely to happen, where gang violence may erupt, or provide context to stop an epidemic. This approach is potentially transformative, but two areas currently limiting a more widespread use are, the need for unwieldy data manipulations in a variety of software packages, and a reliance on internet connectivity. Both these deficiencies limit the uptake of the method in non-spatial disciplines, and with partners working in challenging environments. In this paper, we have addressed these needs in the form of Wordmapper, which allows researchers to effectively process and analyze data being generated in various formats, while also having access to a variety of tools suitable for different spatial skill levels. In so doing, we have now opened the way for SVGs to become a more widely used tool to add context into spatial-social research.

## Figures and Tables

**Figure 1 ijerph-16-00515-f001:**
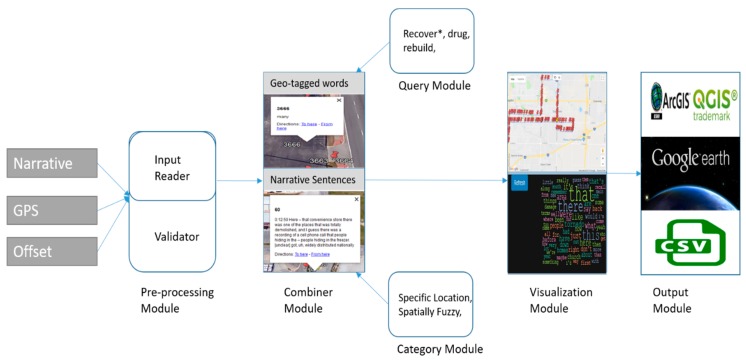
Conceptual diagram for Wordmapper with the six different modules.

**Figure 2 ijerph-16-00515-f002:**
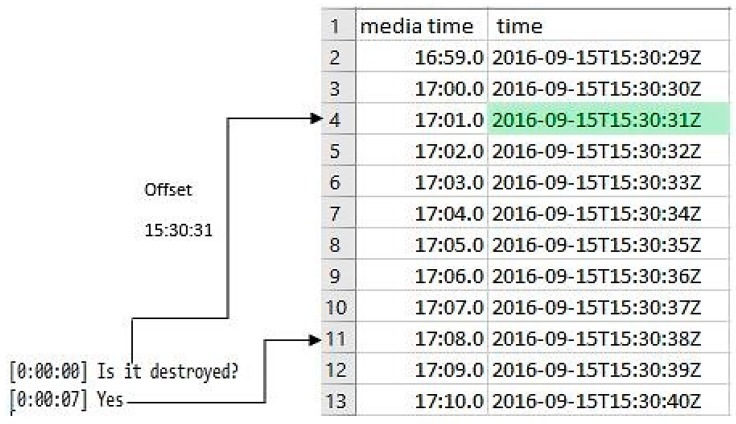
Illustration of the GPS record to narrative mapping.

**Figure 3 ijerph-16-00515-f003:**
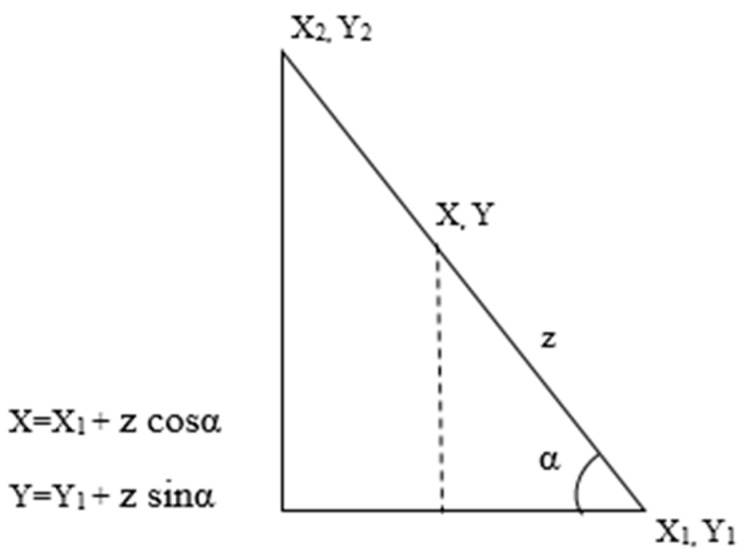
Interpolation algorithm for geotagged words.

**Figure 4 ijerph-16-00515-f004:**
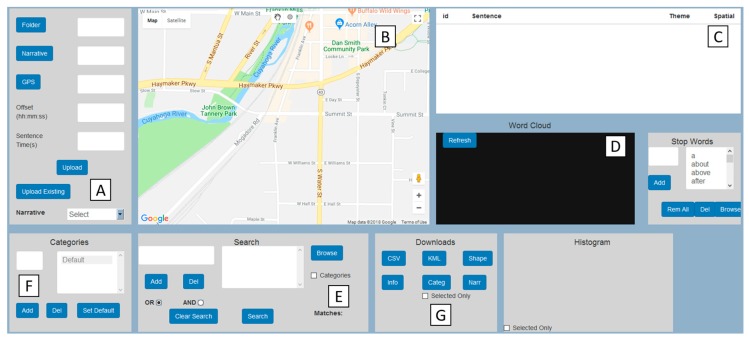
Wordmapper interface diagram (**A**) Input module (**B**) Map in visualization module (**C**) Narrative sentence table (**D**) Wordcloud in visualization module (**E**) Query module (**F**) Category module (**G**) Output module

**Figure 5 ijerph-16-00515-f005:**
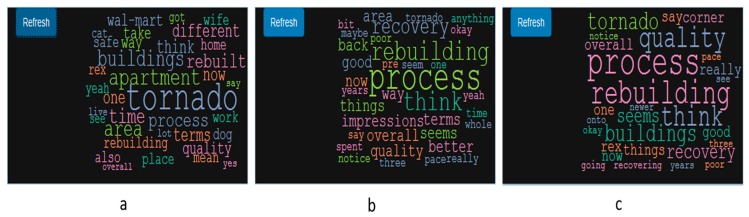
Wordcloud diagram for (**a**) an entire geonarrative (**b**) For the keyword ‘recover*’ (**c**) and then ‘rebuilding’.

**Figure 6 ijerph-16-00515-f006:**
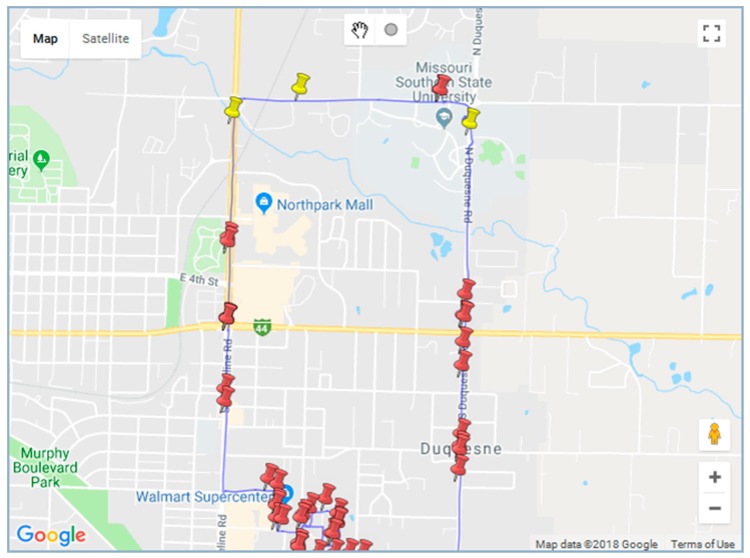
Visualization module showing the narratives containing the searched for keyword ‘recover*’ in yellow pins. All other narrative sentences are shown by red pins.

**Figure 7 ijerph-16-00515-f007:**
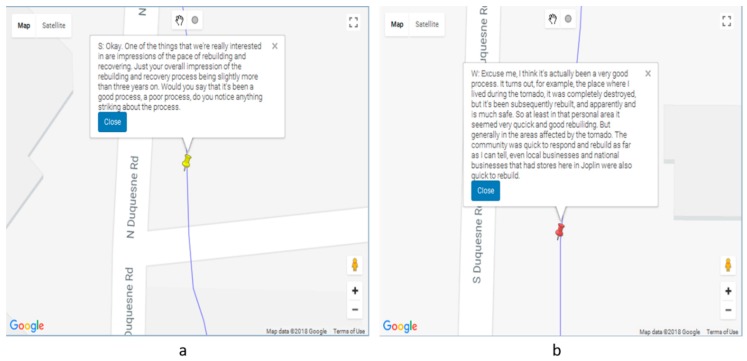
Wordmapper visualization module showing the (**a**) narrative sentence for the keyword ‘recover*’ and (**b**) response by the interviewee about recovery without using the word ‘recover’. The text box in Figure 7a reads, “S: Okay. One of the things that we’re really interested in are impressions of the pace of rebuilding and recovering. Just your overall impression of the rebuilding and recovery process is slightly more than three years on. Would you say that it’s been a good process, a poor process, do you notice anything striking about the process.”. Text box on Figure 7b reads, “W. Excuse me, I think it’s actually been a very good process. It turns out, for example, the place where I lived during the tornado, it was completely destroyed, but it’s been subsequently rebuilt, and apparently and is much safe. So at least in that personal area it seemed very quick and good rebuilding. But generally, in the areas affected by the tornado. The community was quick to respond and rebuild as far as I can tell, even local businesses and national businesses that had stores here in Joplin were also quick to rebuild.”.

**Figure 8 ijerph-16-00515-f008:**
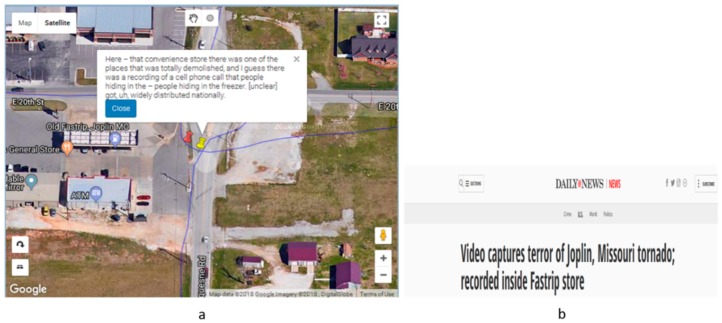
(**a**) Narrative sentence about a convenience store that was completely destroyed in the tornado (**b**) the corresponding tornado article from the Daily News. The text box on Figure 8a reads, “Here – that convenience store there was one of the places that was totally demolished, and I guess there was a recording of a cell phone call that people hiding in the – people hiding in the freezer. [unclear] got, uh, widely distributed nationally.”.

**Figure 9 ijerph-16-00515-f009:**
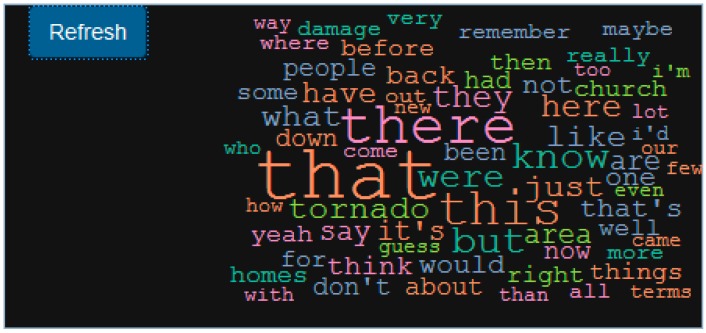
A combined wordcloud for ‘Location specific’ and ‘Fuzzy Space’.

**Figure 10 ijerph-16-00515-f010:**
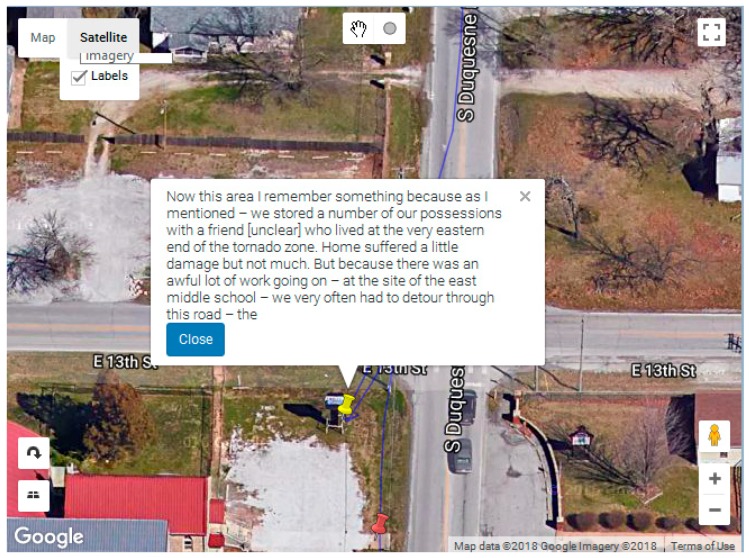
Narrative text for matching keyword ‘damage’ and category ‘Fuzzy space’. The text box on Figure 10 reads, “Now this area I remember something because as I mentioned—we stored a number of our possessions with a friend [unclear] who lived at the very eastern end of the tornado zone. The house suffered a little damage, but not much. But because there was an awful lot of work going on—at the site of the east middle school—we very often had to detour through this road—the”.

**Figure 11 ijerph-16-00515-f011:**
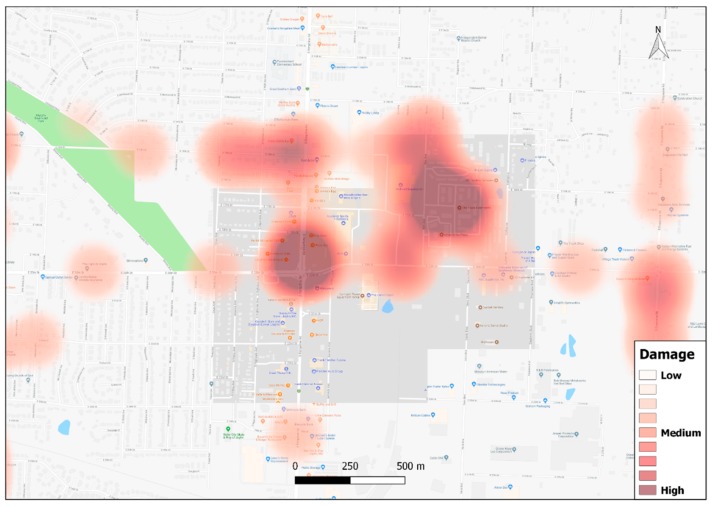
Heat map for ‘damage’ category containing spatial references.

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
