# Peer review of "The Use of Geonarratives to Add Context to Fine Scale Geospatial Research"

_ijerph, 2019, doi:10.3390/ijerph16030515_

Round 1

Reviewer 1 Report

The article describes a software program designed to help non-technical researchers create geonarratives, and performs a compelling case study to demonstrate its potential value for researchers. The authors clearly describe the current gap in solutions for assembling geonarratives, and in doing so provide a strong justification for what they have created. I enjoyed reading this article, and my concerns are quite minor. Hopefully the authors find my comments constructive.

I appreciate the authors’ emphasis on making methods more usable, particularly for researchers without GIS experience. I think we often forget the number of disciplines outside of geography that are adopting GIS (biology, anthropology, health sciences, etc), making accessibility-oriented tools like this desperately needed. Although the authors mention that their tool will help non-GIScientists, I would suggest that they add a few sentences to briefly outline the range of disciplines that have adopted GIS (and therefore number of potential non-GIScientists/users of their application).

The authors mention that the software is meant to work well with limited network access, but given that it uses Google Maps API, a sentence describing whether it works at all offline is warranted. Relatedly, and due to the sensitivity of the data that may frequently be used with this application, also warranted is a sentence or two describing the potential privacy impacts of using the Google Maps API (or perhaps why users shouldn’t worry, if that is the case).

The authors refer to spatial video geonarratives (SVGs) throughout the article. Given that the program as well as the case study don’t seem to make use of video, I’m slightly confused as to why they are given so much emphasis, including in the discussion. I understand their relevance and see why they would be briefly discussed, but keeping more to ‘geonarratives’ would lend to better consistency/clarity.

In the paragraph starting on line 287, the authors claim that their wordcloud and map combination helped to identify that regular people don’t use the word ‘recover’. I’m not sure that I understand how the map helped here; would it not be readily apparent just from the interview transcript alone? Also, ‘empty lots’ and ‘trailers’ are suggested as the next search terms (line 297), but there isn’t any explanation of why these would be used. I assumed these suggestions were based on the results surrounding ‘recovery’, but I couldn’t find mention of ‘empty lots’ or ‘trailers’ terms in Figures 5, 6, 7, or 8.

Finally, I imagine that a mobile app that simultaneously records linked GPS tracks and audio would be very useful, and could be a potential future project for the authors.

Author Response

Thank you for reviewing and providing valuable insights to improve our research. Kindly find the questions and the respective responses below

1) I appreciate the authors’ emphasis on making methods more usable, particularly for researchers without GIS experience. I think we often forget the number of disciplines outside of geography that are adopting GIS (biology, anthropology, health sciences, etc), making accessibility-oriented tools like this desperately needed. Although the authors mention that their tool will help non-GIScientists, I would suggest that they add a few sentences to briefly outline the range of disciplines that have adopted GIS (and therefore number of potential non-GIScientists/users of their application).

Response : We have added a new sentence along with relevant citation to the manuscript emphasizing the relevance of GIS in different disciplines. (Line number 110 to 113)

2) The authors mention that the software is meant to work well with limited network access, but given that it uses Google Maps API, a sentence describing whether it works at all offline is warranted. Relatedly, and due to the sensitivity of the data that may frequently be used with this application, also warranted is a sentence or two describing the potential privacy impacts of using the Google Maps API (or perhaps why users shouldn’t worry, if that is the case).

Response: We have mentioned about the dependency of Google Maps API on internet connection as well as the security concerns created due to the API in the discussion section of the manuscript. (Line number 382 to 390)

3)  The authors refer to spatial video geonarratives (SVGs) throughout the article. Given that the program as well as the case study don’t seem to make use of video, I’m slightly confused as to why they are given so much emphasis, including in the discussion. I understand their relevance and see why they would be briefly discussed, but keeping more to ‘geonarratives’ would lend to better consistency/clarity.

Response: Wordmapper is one of a suite of software’s which have been developed to enhance spatial video geonarrative (SVG). This paper focuses more on textual analysis, but other papers would either use video for, triangulation, help correct GPS, or become a part of video-map-text iterative query processing. However, irrespective of the focus, the method is still the SVG.

4) In the paragraph starting on line 287, the authors claim that their wordcloud and map combination helped to identify that regular people don’t use the word ‘recover’. I’m not sure that I understand how the map helped here; would it not be readily apparent just from the interview transcript alone? Also, ‘empty lots’ and ‘trailers’ are suggested as the next search terms (line 297), but there isn’t any explanation of why these would be used. I assumed these suggestions were based on the results surrounding ‘recovery’, but I couldn’t find mention of ‘empty lots’ or ‘trailers’ terms in Figures 5, 6, 7, or 8.

Response : We have made the relevant modifications in the manuscript to make this more cogent and clear (Line number 296 to 306). 

“Trailers” and “Empty lots” are possibilities that could arise from the extended recovery query (based on previous disaster related experiences) but that in this paper we have not yet followed that particular iterative trail. Subsequent papers are being written that consider the topic of recovery in this setting more fully.

5) Finally, I imagine that a mobile app that simultaneously records linked GPS tracks and audio would be very useful, and could be a potential future project for the authors.

Response : This is definitely one of our future projects and we are already working on it. Thank you for the valuable suggestions.

Reviewer 2 Report

This paper introduces a standalone software, Wordmapper, to process geonarratives from a transcription and GPS paths. I am not an expert of qualitative GIS, but have used GIS in some mixed-method research. I felt this tool would be very useful for data collectors to integrate multimedia information in their field trips. I just have a few minor comments for the authors to consider.

1.       There is already a Word Mapper tool online (https://wordmapper.fas.harvard.edu/). I think the authors should be cautious of using the same brand. Perhaps ‘GeoWordMapper’?

2.       A sophisticated tool needs a post-correction module to correct GPS errors. I don’t know if the preprocessing module deals with GPS coordinates validation and correction.

3.       The authors discussed a lot about SVG, but the software (as shown in Figure 1) is designed to mainly process textual data, rather than video data. Would video data be able to store as BLOB type in the system? Can users play videos?

4.       Figure 3 needs more explanation on the interpolation algorithm.

Author Response

Thank you for reviewing and providing valuable insights to improve our research. Kindly find the questions and the respective responses below

1) There is already a Word Mapper tool online (https://wordmapper.fas.harvard.edu/). I think the authors should be cautious of using the same brand. Perhaps ‘GeoWordMapper’?

Response : The software from Harvard is Word Mapper, which is a web-based tool to support building glossaries using discontinuous textual data. Our software is named Wordmapper (not the exact same), but we would be cautious about any IP violations as you have rightly pointed out.

2) A sophisticated tool needs a post-correction module to correct GPS errors. I don’t know if the preprocessing module deals with GPS coordinates validation and correction.

Response : Necessary edits have been added to the manuscript (Line number: 143 to 146).

3) The authors discussed a lot about SVG, but the software (as shown in Figure 1) is designed to mainly process textual data, rather than video data. Would video data be able to store as BLOB type in the system? Can users play videos?

Response:  Currently Wordmapper tool is developed to extract contextual from the spatial and textual data from SVG. We have been actively developing software solutions that support a combination of video, spatial data, and narratives, which will be freely available for research community in future releases. As a standalone software, video data could be stored as a BLOB type in local databases such as Sqlite3 as you have pointed out, or an upload feature could be supported where the user can locate the video in their file-system.

4) Figure 3 needs more explanation on the interpolation algorithm.

Response: The interpolation section has been expanded to address this concern. (Line number: 169 to 175)